# Advancing Global Pose Refinement: A Linear, Parameter-Free Model for Closed Circuits via Quaternion Interpolation

**DOI:** 10.3390/s24165112

**Published:** 2024-08-07

**Authors:** Rubens Antônio Leite Benevides, Daniel Rodrigues dos Santos, Nadisson Luis Pavan, Luis Augusto Koenig Veiga

**Affiliations:** 1Polytechnic Center, Federal University of Parana, Curitiba 81530-000, PR, Brazil; kngveiga@gmail.com; 2Military Institute of Engineering, Defense Engineering Program, Rio de Janeiro 22290-270, RJ, Brazil; daniel.rodrigues@ime.eb.br; 3Sinop Campus, Mato Grosso State University, Cuiabá 78060-900, MT, Brazil; nadissonluisp@gmail.com

**Keywords:** point clouds, SLERP, Least Squares, SLAM, laser scanning, global pose refinement, pairwise registration

## Abstract

Global pose refinement is a significant challenge within Simultaneous Localization and Mapping (SLAM) frameworks. For LIDAR-based SLAM systems, pose refinement is integral to correcting drift caused by the successive registration of 3D point clouds collected by the sensor. A divergence between the actual and calculated platform paths characterizes this error. In response to this challenge, we propose a linear, parameter-free model that uses a closed circuit for global trajectory corrections. Our model maps rotations to quaternions and uses Spherical Linear Interpolation (SLERP) for transitions between them. The intervals are established by the constraint set by the Least Squares (LS) method on rotation closure and are proportional to the circuit’s size. Translations are globally adjusted in a distinct linear phase. Additionally, we suggest a coarse-to-fine pairwise registration method, integrating Fast Global Registration and Generalized ICP with multiscale sampling and filtering. The proposed approach is tested on three varied datasets of point clouds, including Mobile Laser Scanners and Terrestrial Laser Scanners. These diverse datasets affirm the model’s effectiveness in 3D pose estimation, with substantial pose differences and efficient pose optimization in larger circuits.

## 1. Introduction

Navigating through a three-dimensional world presents significant challenges in fully capturing an object’s surface due to the static nature of the object’s position or the sensor’s perspective. This limitation necessitates either the adjustment of the object’s orientation or altering the sensor’s viewpoint to achieve a comprehensive mapping. Such constraints are the essence of the Simultaneous Localization and Mapping (SLAM) problem. SLAM endeavors to accurately determine the sensor’s position throughout the mapping process, thereby facilitating the simultaneous creation of a detailed global map that integrates multiple viewpoints and observed scenes. The crux of the SLAM challenge lies in the interdependence between accurate positioning and accurate map creation; precise localization on a map requires a correct map, while creating an accurate map depends on precise positioning information.

In practice, SLAM amalgamates a series of computer vision techniques enabling a sensor, like a camera, to self-localize in space while concurrently mapping the environment. Typically, SLAM algorithms comprise two primary components: the front end and the back end. The front end processes sensor data to construct a map, while the back end manages trajectory estimation and localization [1]. A feature descriptor algorithm, often incorporated in the front end, is crucial for accurate displacement calculations [2]. This algorithm should yield discriminative features that remain invariant under rigid 3D transformations, such as translations or rotations. The back-end of SLAM optimizes trajectories using estimated transformations from front-end maps, thereby improving correspondence matching between the maps. Various optimization models, including g2o [3], are utilized in the back-end. It combines Graph Theory with the Levenberg–Marquardt method, an iterative technique encompassing aspects of the Gauss-Newton method and Descending Gradient [4]. Notably, while the front end operates based on the data captured by the sensor, the back end operates solely on poses, making it invariant to the specific data or sensor type used.

This paper presents two novel contributions to the field of LIDAR-SLAM. The first focuses on the LIDAR-SLAM front end, proposing a fully automatic method for registering multiple 3D point clouds using enhanced off-the-shelf methods. This approach facilitates the reconstruction of two Terrestrial Laser Scanner (TLS) datasets from [5] and LiDAR odometry using the Mobile Laser Scanner (MLS) dataset from [6], streamlining the registration process. This eliminates manual intervention and ensures robust, accurate alignment of point clouds.

Our second and more important contribution targets the SLAM back-end, introducing a unique global trajectory optimization technique. This method refines all absolute poses of a sensor, generating a closed linear form without the need for inversions or matrix decompositions. By leveraging this optimization approach, the accuracy and consistency of the sensor’s trajectory estimation are significantly improved, yielding enhanced mapping and localization performance.

## 2. Related Work

Usually, the registration task consists of two modules: the pairwise and the global registration.

### 2.1. Pairwise Registration

Within the field of computer vision, LIDAR technology plays a crucial role in generating precise 3D point clouds, which presents various challenges in terms of mathematical modeling. One such challenge is 3D point cloud registration, which involves aligning two or more point clouds with each other. The goal of registration is to combine multiple sensor readings into a single, coherent, and accurate representation. However, registering point clouds is a complex task due to the inherent ambiguity involved in associating points between different point clouds. As a result, numerous strategies have been proposed to tackle this problem, including global and local methods.

The Iterative Closest Point (ICP) algorithm is a local method that iteratively approximates a pair of point clouds by decomposing their covariance matrix into singular values. The algorithm searches for the closest points between the clouds at each iteration and uses them as pseudo-correspondences to solve the 3D rigid body equations. The ICP algorithm typically employs two stopping criteria: the maximum number of iterations and a threshold for the minimum Root Mean Square Error (RMSE) variation, often set close to zero.

ICP has undergone several improvements since its original idea [7]. For example, the one in [8] changes the minimization function to one that considers the approximation between points and planes, which speeds up convergence. However, convergence to the closest local minimum is common in all ICP variations. Geometrically, local methods are incapable of aligning two clouds with significant variations from the correct position. There are several proposed ICP variations to improve this situation; the most current ones are the Trimmed ICP [9], the Generalized ICP [10], and the Global ICP [11]. Given the many variations of the algorithm, there are also several reviews in the literature; we recommend [12].

Another 3D point cloud registration approach is the global/coarse method, which estimates an approximate transformation for the cloud pair, regardless of the initial orientation. When integrated with ICP variants, these models define the coarse-to-fine registration structure, where a coarse method, which uses little information from the cloud, is used to initialize a refinement method, like ICP variants. Refinement methods use more points, although they are subject to local minima. The advantage of this combination is that the gains of one method outweigh the disadvantages of the other, i.e., coarse methods deliver reliability, while refinement methods deliver accuracy.

One of the most successful global models in the field operates with the descriptor Fast Point Feature Histogram (FPFH), which was introduced by [13], the primary author of the Point Cloud Library (PCL) [14]. The effectiveness of the FPFH descriptor is evident in numerous works in the literature on coarse/global registration methods [2]. For instance, in [15], it was extensively tested against several other descriptors and adopted in their Fast Global Registration (FGR). It is worth mentioning that the authors of FGR are also the main creators of the Open3D library [16], a widely-used Python library for 3D point cloud processing, which we adopt here.

Another common descriptor is the 4-Point Congruent Set (4PCS) [17]. Several works have reviewed the description capability, speed, and robustness of FPFH and 4PCS in different scenarios; [18] compared the descriptors SHOT [19], Spin Image [20], Shape Context [21], and FPFH on actual and multi-source data, and found that FPFH was the most stable and fastest of all. In [22], another comparison involving FPFH is made against 4PCS, Super-4PCS [23,24], which is Open Computer Vision’s registration algorithm. In the authors’ words, the conclusion is that “surprisingly, the FPFH algorithm of Rusu et al. outperforms all other approaches, including the most recent ones”.

Emerging advancements in the field have presented new algorithms that claim superior performance in terms of accuracy and speed compared to the FPFH descriptor. Notably, TEASER [25] and the Binary Shape Context (BSC) descriptor [26] have demonstrated promising results. Furthermore, a novel category of algorithms based on neural networks, specifically leveraging the PointNet [27] architecture, has recently emerged. While originally designed for semantic segmentation of point clouds, it has been recognized that object orientation plays a significant role in 3D recognition. The varying poses of objects introduce distinct features, enabling object pose prediction to aid in class label prediction [28]. Given that point cloud registration inherently aims to predict the pose of the cloud, some studies [29] use PointNet-based neural networks for pairwise registration tasks.

### 2.2. Global Refinement

Global refinement models exist to mitigate the drift that arises after the sequential registration of multiple pairs of 3D point clouds on a dataset. For TLS-based point cloud datasets, this is usually a small problem, as there are usually few clouds. However, this is a fundamental step to make the sensor trajectory consistent and free of drift for MLS and RGB-D datasets.

An ambiguity arises in the 3D point cloud registration literature regarding global registration. Some authors [15] employ this term to denote coarse pairwise registration, where global implies freedom from local minima. However, within the same context, in Ref. [5] global refers to algorithms that leverage all transformations between point clouds in a given dataset. These algorithms aim to optimize the trajectory by utilizing the poses obtained from pairwise registration to address the drift error. We call them the Global Refinement Model (GRM) to avoid confusion with global registration. 

Usually, a GRM represents poses using graphs g(v,e), where each vertex v corresponds to a sensor location, and each edge e connecting two vertices represents a relative pose. However, optimizing these graphs presents a highly nonlinear problem [30,31,32] due to the vast parameter space involved in simultaneous pose optimization. Furthermore, the determination of overlapping pairs of point clouds is inherently uncertain. Consequently, specific models, like those introduced by [5,33], resorting to exhaustively testing all possible pairs within the dataset, could be inefficient for larger datasets.

When dealing with MLS data, multiple overlaps between nearby point-cloud pairs are expected to be considered. However, the scenario differs substantially for TLS-derived point cloud pairs, as they are typically more spaced apart with fewer locations. In this context, to mitigate the challenges posed by the vast search space in complete graphs, Ref. [34] employed global descriptors to detect overlapping point clouds, constructing a hierarchical graph based on cloud similarity. The BSC [26] descriptor performs pairwise registration, and the GRM adopted for drift correction is the one of [35]. In [36], they focused on plane correspondences for registration, employing Singular Value Decomposition (SVD) to optimize rotations and the method proposed by [32] for global fitting of translations.

For RGB-D data, [22] exhaustively registered point cloud pairs using the FPFH [13], and the GRM for drift correction is the g2o framework of [3] applied in a complete graph. Working with MLS data, Ref. [37] performed pairwise registration using the 4PCS [17] and achieved drift correction through an GRM based on Spherical Linear Interpolation (SLERP) of rotations mapped in quaternions [38]. Optimizing rotations also has the benefit of improving the translations of the poses from the trajectory. In the following sections, we will describe how our method integrates several well-established models to achieve a fast and reliable reconstruction of 3D point cloud datasets without needing a complete graph.

## 3. Method

To address the objective of registering multiple pairs of 3D point clouds without drift, we propose a pairwise coarse-to-fine registration approach that combines the FGR [15] and Generalized Iterative Closest Point (GICP) [10] in a multiscale manner. With the relative poses returned by the pairwise registration, we feed the proposed global refinement method, which will return globally refined poses. This flow chart is in Figure 1.

### 3.1. Pairwise Registration

Figure 2 illustrates the two-step process of our proposed point cloud registration approach: the coarse registration step (a) and the refinement phase via Multiscale Generalized Iterative Closest Point (M-GICP) (b). For a given point cloud pair consisting of reference and target clouds, the process begins with voxelization for downsampling, followed by Statistical Outlier Removal (SOR) filtering, and then estimating the normals of the points. These preliminary steps are vital in enhancing the registration function’s performance, accounting for cloud density differences (via voxelization), reducing local minima (through filtering), and preparing the data for the Fast Point Feature Histogram (FPFH) descriptor estimation in the Fast Global Registration (FGR) step, which requires point normals from both clouds.

As displayed in Figure 2b, our method enhances the robustness of the GICP algorithm by implementing it in a multiscale framework that employs decreasing voxel sizes to progressively downsample the point cloud pair. Additionally, we apply a robust weight function to the correspondences determined by the M-GICP using L1 metric penalization. This weight function, which requires no parameters, has demonstrated commendable performance across various datasets, as supported by [39].

The M-GICP approach represents a multiscale variant of the GICP algorithm, where the output of the FGR method initializes the registration at the coarsest scale. It functions by iteratively registering the point cloud pair at increasingly refined scales, initiating from a relatively larger voxel size that considerably downsamples the point clouds. Utilizing larger voxels at coarser scales enables efficient convergence, particularly for voluminous point clouds, and helps smooth out local minima in the registration function. The subsequent finer scales employ smaller voxel sizes, initializing the GICP using the registration result obtained from the preceding scale. This multiscale strategy promotes accelerated convergence by leveraging a reduced point set at coarser scales, all while preserving high accuracy at finer scales. In turn, it balances efficiency and precision, simultaneously boosting the GICP’s robustness against initial orientation variations in the point cloud pair.

FGR and multiscale GICP use several parameters to register a point cloud pair. After their presentation in Table 1, we discuss each in the following sections.

#### 3.1.1. Voxel Sizes

In the 3D processing literature, voxels or Volume Elements, are subdivisions of 3D space into octants. Like pixels, voxels can encompass various features, including the centroid of points within the voxel for 3D point clouds. Replacing original points with voxel centroids can significantly reduce point cloud size while retaining geometric information. Additionally, it allows for a more uniform distribution of points, benefiting subsequent steps such as filtering, normal estimation, and registration.

The optimal voxel size is a critical variable influenced by the intended use of the point cloud and the specific data acquisition sensor. Prior studies [5,32] have suggested a voxel size of 0.1 m for TLS-derived point clouds, reducing the point cloud size by two orders of magnitude while maintaining necessary geometric information. However, MLS-derived point clouds typically feature lower point density and overall point count. Despite this, many collected clouds necessitate downsampling for algorithm performance maintenance. As per the specifications in Table 1, we implement voxel sizes of 0.1 m for the FGR model in the coarse registration stage and a sequence of four progressively decreasing voxel sizes for the GICP iterations. This multiscale approach defines four unique scales for the point clouds, enhancing the speed and robustness of the Generalized Iterative Closest Point (GICP) registration process.

#### 3.1.2. Maximum Correspondence Distances

The matching distance parameter determines the maximum allowable distance for establishing correspondences between source and target cloud points. Influenced by the sampling voxel size and the data’s characteristics, larger matching distances are necessary for TLS-derived point clouds due to abrupt pose changes between stations. In contrast, smaller values below 0.5 m are typically adequate for sequentially collected MLS-acquired point clouds. The matching distance should ideally be two to four times the sampling voxel size. Larger values increase the points considered in the registration loop but with a higher false-positive rate. Conversely, smaller values restrict points in the registration process, reducing the false-positive rate but increasing the risk of misregistration for substantial pose differences. Given our multiscale approach, a single multiplier is used for all voxel sizes within the GICP algorithm. It should be noted that the Fast Global Registration (FGR) stage only uses this distance parameter to calculate the Root Mean Square Error (RMSE) registration metric as it does not rely on neighboring search to establish point correspondences.

#### 3.1.3. Neighborhood of the Multiscale SOR Filter

At each sampling scale, the cloud pair undergoes successive filtering using the Statistical Outlier Removal (SOR) filter [40]. This filter depends on two key parameters: the knn number and the scalar α that scales the standard deviation. The filter analyzes each point in the cloud using its k-nearest-neighbors (knn). Points with a mean distance to neighbors outside the range defined by [−ασ, ασ] are considered outliers and removed, with σ representing the standard deviation and α the multiplier that controls the intensity of the filter. Initially, a more aggressive (small α) and precise (high knn) filter configuration is employed to leave only the macro-structures of the point cloud. As the registration progresses to finer scales, the knn value is halved at each scale to compensate for the exponential increase in the number of points in the cloud. On the other hand, α values become higher to allow microstructures of the point cloud to appear. This strategy facilitates efficient filtering across scales and allows previously discarded information to be used by the registration.

#### 3.1.4. Neighborhood for Normal and Covariance Estimation

Though important, this neighborhood parameter is less critical to the registration process. Our approach uses the Fast Library for Approximate Nearest Neighbors [41] (FLANN) to estimate normals via a hybrid search method. This method incorporates a radius and a maximum number of neighbors as criteria. When the number of points within the specified radius exceeds the maximum K-nearest neighbors (KNN) value, the estimation is limited to the maximum KNN value. Normal estimation is performed using the Singular Value Decomposition (SVD) method, with the smallest eigenvalue’s eigenvector from the decomposition representing the desired normal vector. We set a maximum KNN value of 20 and utilize twice the voxel size of the current scale for the radius for normal and covariance matrix estimation. Alterations to these parameter values have minimal impact on the registration results and overall accuracy.

#### 3.1.5. Weight Function

The weighing stage is just as fundamental as filtering because a large enough outlier can corrupt the entire result of the Least Squares (LS) estimate. The weighing of matches in ICP dates back to the testing framework in [42] and, more recently, to [43], which remain standard in ICP step sequences. These steps consist of (1) sampling, (2) filtering, (3) associating matches, (4) weighing the matches, and (5) minimizing the error function. Weighting can be understood as filtering when it acts on the correspondences by assigning a weight of 0 to some pairs; in this case, it is configured as hard-reject and can have a variable maximum error threshold, as occurs in Trimmed-ICP.

Defining a continuous weight function for all observations leads to an LS model with iterative reweighting for the ICP. The most recent and complete analysis of weight functions for ICP, by [39], tested 11 weight functions with real data in indoor and outdoor scenarios. Among these, only two functions, L1 and L2, do not require an error threshold k, with L2 representing uniform weighting, assigning a weight w=1 for all pairs. Conversely, certain functions, categorized as M-estimators, necessitate the adjustment of k, potentially yielding inferior outcomes compared to uniform weighting. As an example of this phenomenon, Ref. [39] mentions the Geman-McClure function, which is used in the FGR.

Consequently, for the M-GICP, we have chosen the L1 weight function, which assigns a weight w=1/e, with $e=\left|Tp_{i}-q_{j}\right|$, where $p_{i}\in P$ and $q_{j}\in Q$. P and Q are the source and target point cloud; and T is the current GICP transformation that maps point $p_{i}$ to the system of $q_{j}$. This decision is particularly pertinent given the challenge of defining k within a multiscale context, where its optimal value is highly contingent upon the point cloud’s sampling density. Furthermore, Babin’s analysis underscores the L1 function’s robust performance.

### 3.2. Proposed Global Refinement Model

The proposed GRM operates in a closed loop, consisting of two separate stages: the refinement of global rotations and global translations. Both optimizations are independent and do not require any iteration, matrix inversion, or decomposition. That is, both are linear and deduced in a closed form.

We will now deduce our method for correcting the rotations of the pose circuit. To derive a global origin from a closed loop of relative poses in the sensor pose graph, the pose rotation matrices are first transformed into unit quaternions, denoted as $\left|q_{i}\right|=1$, and then combined through the following sequence of multiplications:(1)$$\begin{array}{c} r_{0}={q_{0}=q}_{I}\\ r_{1}=q_{1}r_{0}\\ \begin{array}{c} \vdots \\ r_{n}=q_{n}r_{n-1}\\ r_{e}=q_{lc}r_{n} \end{array} \end{array}$$
the rotation of the pose $x_{i}$ relative to the pose $x_{0}$ is represented by the quaternion called $r_{i}$ (global rotations); $q_{i}$ represents the rotation of the pose $x_{i}$ relative to the pose $x_{i-1}$ (relative rotations), and $q_{I}$ represents the identity rotation as the quaternion (1,0,0,0). This applies to all poses from i=0 to n. $q_{lc}$ is the loop-closure quaternion, the rotation which closes the circuit, it references pose $x_{0}$ at pose $x_{n}$; and $r_{e}$ is the rotation closure error. If there were no errors in the rotation estimates, the quaternion $r_{e}$ would be the identity quaternion $q_{I}$. However, in practice, it approximates that ($r_{e}≅q_{I}$). By composing the rotations of the poses $x_{n}$, $x_{n-1}$, …, $x_{1}$ in the opposite direction of the sensor’s trajectory, we can obtain the rotations of these poses relative to the origin $x_{0}$ in another way:(2)$$\begin{array}{c} p_{1}={\left(q_{2}\ldots q_{n}q_{lc}\right)}^{-1}\\ p_{2}={\left(q_{3}\ldots q_{n}q_{lc}\right)}^{-1}\\ \begin{array}{c} \vdots \\ p_{n}={\left(q_{lc}\right)}^{-1} \end{array} \end{array}$$
the multiplication properties of quaternions allow us to rewrite $p_{i}$ as:(3)$$\begin{array}{c} p_{1}=r_{1}r_{e}^{-1}\\ p_{2}=r_{2}r_{e}^{-1}\\ \begin{array}{c} \vdots \\ p_{n}=r_{n}r_{e}^{-1} \end{array} \end{array}$$
to achieve an optimal rotation based on the quaternions $r_{i}$ and $p_{i}$, which represent the same rotation of the pose $x_{i}$ referenced in the pose $x_{0}$, with i = 1, 2, …, n, we demonstrate that it is better to interpolate $r_{i}$ and $p_{i}$ using the SLERP technique in the following manner:(4)$$\begin{array}{c} {\hat{q}}_{1}=r_{1}{\left(r_{1}^{-1}p_{1}\right)}^{\tau_{1}}\\ {\hat{q}}_{2}=r_{2}{\left(r_{2}^{-1}p_{2}\right)}^{\tau_{2}}\\ \begin{array}{c} \vdots \\ {\hat{q}}_{n}=r_{n}{\left(r_{n}^{-1}p_{n}\right)}^{\tau_{n}} \end{array} \end{array}$$
herein ${\hat{q}}_{i}$ is the interpolated quaternion between $r_{i}$ and $p_{i}$, at the interval $\tau_{i}\;\in \left[0,1\right]$, representing the rotation of vertex $x_{i}$ referenced to vertex $x_{0}$, with i = 1, 2, …, n. Note that the rotation $r_{0}$ of the first pose is not present, as it is fixed in the global refinement. Substituting Equation (3) in Equation (4), we have:(5)$$\begin{array}{c} {\hat{q}}_{1}=r_{1}{\left(r_{1}^{-1}r_{1}r_{e}^{-1}\right)}^{\tau_{1}}=r_{1}r_{e}^{-\tau_{1}}\\ {\hat{q}}_{2}=r_{2}{\left(r_{2}^{-1}r_{2}r_{e}^{-1}\right)}^{\tau_{2}}=r_{2}r_{e}^{-\tau_{2}}\\ \begin{array}{c} \vdots \\ {\hat{q}}_{n}=r_{n}{\left(r_{n}^{-1}r_{n}r_{e}^{-1}\right)}^{\tau_{n}}=r_{n}r_{e}^{-\tau_{n}} \end{array} \end{array}$$
to optimize SLERP intervals $\tau_{i}$, with i = 1, 2, …, n, one takes advantage of the constraint that the triple composition ${\hat{q}}_{i}^{-1}q_{i}{\hat{q}}_{\left(i-1\right)}$, must approximate as much as possible to the identity quaternion, as follows:(6)$$\begin{array}{c} {\hat{q}}_{1}^{-1}q_{1}{\hat{q}}_{0}^{-1}={\left(r_{1}r_{e}^{-\tau_{1}}\right)}^{-1}q_{1}\approx q_{I}\\ {\hat{q}}_{2}^{-1}q_{2}{\hat{q}}_{1}={\left(r_{2}r_{e}^{-\tau_{2}}\right)}^{-1}q_{2}r_{1}r_{e}^{-\tau_{1}}\approx q_{I}\\ \begin{array}{c} \vdots \\ {\hat{q}}_{n}^{-1}q_{n}{\hat{q}}_{n-1}={\left(r_{n}r_{e}^{-\tau_{n}}\right)}^{-1}q_{n}r_{n-1}r_{e}^{-\tau_{n-1}}\approx q_{I}\\ q_{lc}{\hat{q}}_{n}=q_{lc}r_{n}r_{e}^{-\tau_{n}}\approx q_{I} \end{array} \end{array}$$
note that ${\hat{q}}_{0}^{-1}$ equals $q_{I}$. Performing the appropriate algebra of quaternions and replacing Equation (1) in Equation (6) ($r_{i}=q_{i}r_{i-1}$), the above equations are rewritten as:(7)$$\begin{array}{c} {\hat{q}}_{1}^{-1}q_{1}{\hat{q}}_{0}^{-1}=r_{e}^{\tau_{1}}r_{1}^{-1}r_{1}=r_{e}^{\tau_{1}}\approx q_{I}\\ {\hat{q}}_{2}^{-1}q_{2}{\hat{q}}_{1}=r_{e}^{\tau_{2}}r_{2}^{-1}r_{2}r_{e}^{-\tau_{1}}=r_{e}^{\tau_{2}}r_{e}^{-\tau_{1}}\approx q_{I}\\ \begin{array}{c} \vdots \\ {\hat{q}}_{n}^{-1}q_{n}{\hat{q}}_{n-1}=r_{e}^{\tau_{n}}r_{n}^{-1}r_{n}r_{e}^{-\tau_{n-1}}=r_{e}^{\tau_{n}}r_{e}^{-\tau_{n-1}}\approx q_{I}\\ q_{lc}{\hat{q}}_{n}=r_{e}r_{e}^{-\tau_{n}}\approx q_{I} \end{array} \end{array}$$
through the property of multiplying powers of quaternions, $q^{\tau_{1}}q^{\tau_{2}}=q^{\tau_{1}+\tau_{2}}$, which applies to equal quaternions, we can deduce that:(8)$$\begin{array}{c} {\hat{q}}_{1}^{-1}q_{1}{\hat{q}}_{0}^{-1}=r_{e}^{\tau_{1}}\approx q_{I}\\ {\hat{q}}_{2}^{-1}q_{2}{\hat{q}}_{1}=r_{e}^{\tau_{2}}r_{e}^{-\tau_{1}}=r_{e}^{\tau_{2}-\tau_{1}}\approx q_{I}\\ \begin{array}{c} \vdots \\ {\hat{q}}_{n}^{-1}q_{n}{\hat{q}}_{n-1}=r_{e}^{\tau_{n}}r_{e}^{-\tau_{n-1}}=r_{e}^{\tau_{n}-\tau_{n-1}}\approx q_{I}\\ q_{lc}{\hat{q}}_{n}=r_{e}r_{e}^{-\tau_{n}}=r_{e}^{1-\tau_{n}}\approx q_{I} \end{array} \end{array}$$
to ensure that every quaternion raised to 0 equals the identity, we need to minimize each power of Equation (8) to obtain the best intervals of $\tau_{i}$. We do this by minimizing the following linear system:(9)$$\begin{array}{c} \tau_{1}=0\\ \tau_{2}-\tau_{1}=0\\ \begin{array}{c} \vdots \\ \tau_{n}-\tau_{n-1}=0\\ 1-\tau_{n}=1 \end{array} \end{array}$$
for this task we use the LS method:(10)$$\hat{\tau }={\left(J^{T}J\right)}^{-1}J^{T}b$$
where J is the Jacobian matrix of Equation (9) concerning variables $\tau_{1},\tau_{2},\ldots ,\tau_{n}$; b is the vector of desired values for the combination of exponents in Equation (9) $b=\left[0,0,\ldots ,1\right]$, which comes directly from the exponents in (8); and $\hat{\tau }$ is the desired vector of interpolation intervals. As the result of Equation (10) is a function of the number of stations, for a closed circuit of n vertices, $\hat{\tau }$ is previously known, since the matrix J and the vector b follow a simple pattern:(11)$$J={\left[\begin{array}{cc} \begin{array}{ccc} 1 & 0 & \ldots \\ -1 & 1 & \ldots \\ \vdots & \vdots & \ddots \end{array} & \begin{array}{cc} 0 & 0\\ 0 & 0\\ \vdots & \vdots \end{array}\\ \begin{array}{ccc} 0 & 0 & \ldots \\ 0 & 0 & \ldots \end{array} & \begin{array}{cc} -1 & 1\\ 0 & -1 \end{array} \end{array}\right]}_{\left(n,n-1\right)}\;;\;b={\left[\begin{array}{c} 0\\ \begin{array}{c} 0\\ \vdots \\ 0 \end{array}\\ 1 \end{array}\right]}_{n-1,1}$$
for example, in a circuit with n = 5 vertex, the vector $\hat{\tau }$ contains n−1 values, as the first pose is not refined:(12)$$\hat{\tau }=\left[\begin{array}{cccc} \frac{1}{5} & \frac{2}{5} & \frac{3}{5} & \frac{4}{5} \end{array}\right]$$
i.e., we can find SLERP intervals using:(13)$$\frac{1}{n},\;\ldots ,\frac{n-2}{n},\frac{\left(\;n-1\right)}{n}$$

LS solution reveals that the interval [0, 1] is divided into n−1 linear parts. To put it simply, the SLERP interpolation is the optimal estimate between $r_{i}$ and $p_{i}$. Since $r_{i}$ and $p_{i}$ accumulate different amounts of rotations, the interpolation intervals $\tau_{i}$ are linearly proportional to this value. It is worth noting that in a circuit with an even number of poses, one of the pairs of quaternions will be interpolated exactly by ½ since going in the reverse or forward direction accumulates the same number of rotations, therefore the same amount of error. A homogeneous error distribution along the circuit is assumed, which is not strictly true but tends to be as the number of poses in the circuit grows. In a sense, a linearly increasing distribution of the rotation error in the poses is considered.

Once the rotations have been refined, we turn our focus to the refinement of translations, employing the linear model of Lu and Milios (let us call this model LUM) [32]. The configuration of the Jacobian hypermatrix is shaped by the alignment of poses within the graph, encapsulating a simple closed-loop topology. If we continue with our prior example involving a loop comprising five poses, we will observe the following structure for the hypermatrix H:(14)$$H=\left[\begin{array}{cc} \begin{array}{cc} I & 0\\ -I & I\\ 0 & -I \end{array} & \begin{array}{cc} 0 & 0\\ 0 & 0\\ I & 0 \end{array}\\ \begin{array}{cc} 0 & 0\\ 0 & 0 \end{array} & \begin{array}{cc} -I & I\\ 0 & -I \end{array} \end{array}\right]\;I=\left[\begin{array}{ccc} 1 & 0 & 0\\ 0 & 1 & 0\\ 0 & 0 & 1 \end{array}\right]$$
the vector of observations is organized as follows:(15)$$d=\left[\begin{array}{c} R_{I}t_{10}\\ {\hat{R}}_{10}t_{21}\\ \begin{array}{c} {\hat{R}}_{20}t_{32}\\ {\hat{R}}_{30}t_{43}\\ {\hat{R}}_{40}t_{04} \end{array} \end{array}\right]$$
${\hat{R}}_{i,0}$ is the rotation matrix obtained from the interpolated quaternion ${\hat{q}}_{i}$, and $t_{\left(i,i-1\right)}$ are the relative translations obtained by registration of point cloud pairs. The LS solution is given as:(16)$$\hat{t}={\left(H^{T}H\right)}^{-1}(H^{T}d)$$
where $\hat{t}$ is the vector of refined translations.

## 4. Experiments and Discussion

We evaluate the performance of our proposed model using three datasets encompassing different sensing modalities. Specifically, we utilize two datasets captured with TLS named Facade and Courtyard, alongside one dataset obtained from an MLS acquired with a Velodyne^®^ sensor. The TLS datasets were originally collected by [5] in an outdoor environment. Each dataset comprises a varying number of point clouds, specifically seven clouds for the Facade and eight clouds for the Courtyard. The point clouds obtained from TLS exhibit high point density, with each point cloud containing over 20 million points. Notably, the point cloud density varies due to the angular acquisition mode employed by the TLS. The Facade dataset represents an urban environment characterized by diverse elements such as facades, vegetation, and dynamic objects caused by the movement of vehicles and pedestrians. The average overlap between the point clouds in the Facade dataset is approximately 60%. Conversely, the Courtyard dataset depicts a desert scenario without vertical structures, exhibiting an average overlap of approximately 70%. The primary objective of utilizing these two TLS datasets is to assess the efficacy of our proposed pairwise registration method. The third dataset is a North Campus Long Term (NCLT) dataset [6] sample. It is a 645 m closed circuit with 901-point clouds obtained by MLS. The average overlap between clouds is 70% and can vary to less than 30% in curves. Each cloud has about 30,000 points and comprises streets with vegetation, cars, buildings, and moving artifacts. Figure 3 presents the described datasets: Facade, Courtyard, and NCLT circuit. 

The primary purpose of this dataset is to test our GRM’s ability to distribute the drift in the circuit through loop closure correction. Scripts written in Python using the Open3D library are available in a repository on GitHub to manipulate all clouds. The authors’ hardware consists of an Intel i3-9400KF CPU (4.6 GHz) with four cores and 16 GB of memory, without a GPU. 

We assess the success rate of our proposed coarse-to-fine registration, FGR+M-GICP, utilizing all potential combinations within the Facade and Courtyard datasets, totaling 49 point cloud pairs. Regarding the NCLT dataset, we limit our evaluation to the global trajectory correction model, given that registering point cloud pairs derived from MLS is relatively easy. Conversely, our proposed SLERP+LUM model for drift correction undergoes assessment solely on the NCLT circuit. This selective evaluation is due to difficulty detecting drift within smaller circuits, such as those produced by TLS clouds.

For the SLERP+LUM evaluation, we measure the translation error of all poses relative to the groundtruth poses of the dataset. The translation error is given by ${\varepsilon t}_{i}=\left\|t_{i}^{g}-t_{i}\right\|$. The superscript g refers to the groundtruth pose of the respective pair and i ranges from 1 to the nth absolute pose. While the SLERP technique directly improves the rotations of the poses, we chose not to measure the error in the rotation because it directly influences the translations of the absolute poses. If we were to do the opposite, comparing our model against the LUM model, for instance, would be unfeasible. This is because translation modifications do not impact rotations, whereas the reverse does have an effect.

### 4.1. Pairwise Registration of the Dataset Courtyard

We establish a set error threshold for translation and rotation to ascertain the accuracy of the pair registration. The registration is deemed successful if the derived error concerning the groundtruth pose remains within these thresholds. The FGR+M-GICP correctly registered all pairs in the Courtyard dataset. The visual results for each pair are shown in Figure 4, the RMSE is in Figure 5, and the time analyses are in Figure 6.

As depicted in Figure 4, all point cloud pairs in the Courtyard dataset have been successfully registered. This dataset illustrates a complete graph of 28 transformations, given that all point cloud pairs overlap. The poses were notably close to the ground truth, and the Root Mean Square Error (RMSE) for all inlier correspondences within the overlapping regions of all pairs remained low, fluctuating between 4 and 7 cm. This result is quite consistent, especially when compared to the 10 cm downsampling voxel utilized in the final scale of M-GCIP registration. Figure 6 shows a reasonable running time for the pairwise registration algorithm, given the complexity of the scene. On average, registration times were under 170 s.

### 4.2. Pairwise Registration of the Dataset Facade

Now, we proceed to the analysis of the second dataset, Facade, which has seven point clouds and 21 possible pairs, as all point clouds overlap with each other. The visual results for each pair are shown in Figure 7, the RMSE is shown in Figure 8, and the time analyses are shown in Figure 9.

Figure 7 demonstrates the successful registration of all point cloud pairs using our proposed FGR+M-GICP algorithm. The effectiveness of this method primarily stems from the FGR, which is especially adept at globally registering distant pairs, such as those obtained by TLS. Figure 8 shows that the RMSE is acceptable within the degree of cloud sampling; this RMSE is calculated only in the overlap area and using a maximum correspondence distance of twice the size of the downsampling voxel, which was 10 cm. Figure 9 shows a reasonable running time for the pairwise registration algorithm, given the complexity of the scene. On average, registration times were under 30 s. Figure 10A shows how M-GICP offers an additional correction capability, effectively handling even significant translation errors. We compare the Vanilla GICP and M-GICP in Figure 10A,C. Figure 10B illustrates the multiscale voxel downsampling, while Figure 10C plots the RMSE against iterations over five scales.

The significant reductions in RMSE, as illustrated in Figure 10C, are attributed to the multiscaling technique. This approach enables a substantial initial search distance for matching points between clouds at the beginning scales, where the clouds are more sparsely distributed. We initialized both algorithms with identical parameters, including the maximum number of iterations and the minimum RMSE variation. However, achieving a correct configuration with the Vanilla version is not feasible due to its fixed, low search distance. The graph in Figure 10C demonstrates that a lower RMSE does not always correlate with successful registration. This is because GICP Vanilla operates on the dense cloud pair and calculates RMSE within the overlap area. In summary, our multiscale approach outperforms the vanilla version in accuracy and speed. The initial scales, being sparse, facilitate faster processing and large movements, while the subsequent denser scales ensure greater accuracy.

### 4.3. Global Refinement Model SLERP+LUM

We will now examine the performance of our SLERP+LUM model using the NCLT dataset, comprising 901 sequentially obtained cloud pairs with variable overlaps ranging from 90% to 30% between pairs. The pairs featuring minimal overlap occur in curves and narrower areas. Due to practicality, we do not visually inspect each pair. Instead, we analyze the differences between the groundtruth poses and those refined by the models. Our primary focus is to assess the impact of global correction, which is shown mainly in the reduction of drift along the circuit. Therefore, we compare the circuit after each GRM. Showcasing the effects of global optimization, Figure 11 displays the circuit reconstructed using LUM in Figure 11A, SLERP in Figure 11B, g2o in Figure 11C and SLERP+LUM in Figure 11D.

In Figure 11A, depicting the LUM model, a notable outward bulging characterizes the circuit’s side view trajectory. Conversely, the SLERP model, as shown in Figure 11B, markedly reduces this bulging. However, it introduces a significant gap between the trajectory’s start and end, clearly observable by the abrupt color shift from green to blue on the right-hand side. Our SLERP+LUM model, illustrated in Figure 11C, successfully closes the circuit and exhibits the smallest degree of bulging in the side view. It aligns the start and end of the trajectory, resulting in a more level and consistent route. Our model could not correct all the drift in the NCLT dataset. Still, it is superior to the previous ones. In the case of the g2o model presented in Figure 11D, the trajectory stabilizes in the lower section for approximately one-third of the circuit’s total length. Nevertheless, the circuit twists significantly in the upper section, which is evident in the side view and through the color variation in the gradient. In Figure 12, we show the error of each model relative to the groundtruth. 

Figure 12 shows a considerable rise in pose error as the distance from the origin increases, primarily due to errors between point cloud pairs. To achieve better results, adding more clouds to the dataset is necessary. However, it is crucial to remember that our objective is not to create a consistent, error-free circuit but rather to compare models under challenging conditions. In practice, spanning 640 m with 901 point-clouds leads to significant distances between pairs, a deliberate design to highlight drift. The increase in error does not follow a specific pattern but often reaches a peak around the midpoint of the circuit, around pose number 450. All the models, except g2o, show a downward error trend towards the end of the circuit, which indicates their circuit-closing nature. The total error in each trajectory was as follows: LUM = 14,707 m; SLERP = 16,901 m; SLERP+LUM = 11,697 m; g2o = 7105 m. Although the quantitative analysis shows superior performance of the g2o model, with this GRM, the trajectory suffers a considerable increase in error near pose 325, probably because in this part of the circuit, the platform that captures the point clouds enters a narrow corridor, losing most of the geometric information and thus recording pairs of point clouds unreliably. Figure 13 shows all the trajectories compared with the ground truth.

Figure 13 highlights the geometric disparities between the GRMs. Initially, the g2o model remains closest to the true trajectory. However, it deviates drastically from the actual path after passing through the lower left corner, an area characterized by low overlap between the circuit’s clouds. When comparing the LUM and SLERP models, the latter has a slight advantage, especially at the end of the circuit. It is also clear that our combination, SLERP+LUM, is better than the individual application of these models, but the same cannot be said compared to the g2o model. The g2o model would have performed better if it had not suffered an abrupt change in the specific pose with low overlap. The proximity of the model to the groundtruth in the first 1/3 of the trajectory is evidence of this. However, we visually analyzed the pairwise registration of each pair in the region of abrupt change and did not find gross errors in the clouds. Therefore, we consider the comparison fair to the point of stating that our model surpasses the g2o results, at least qualitatively, as shown in Figure 11. In Figure 14, we show the temporal analysis of each GRM.

The graph in Figure 14 illustrates that, although the differences in execution time are modest, they remain crucial, particularly since MLS point cloud circuits encompass thousands of cloud pairs, necessitating rapid processing for SLAM operations on various platforms. Temporal analysis reveals that the LUM model is the quickest, succeeded by the SLERP, SLERP+LUM, and then the g2o model. The speed of the LUM model and the SLERP model is mainly because they do not use iterations. Conversely, the g2o model employs the Levenberg–Marquardt method, involving complex processes such as matrix inversions, handling information matrices, iterations, and parameter settings. In addition to being faster, the SLERP+LUM model has a more stable time of execution than the g2o model, but the g2o model holds an advantage over our GRM, as it can navigate graphs with more intricate topologies, including multiple connections between non-adjacent clouds. On the other hand, our GRM is limited to closed circuits, making its adaptation to multi-edge graphs a complex challenge. Similarly, the LUM model functions linearly in a closed loop but, akin to g2o, it can extend to graphs with multiple edges. When this expansion occurs, the model abandons its linear nature and demands intricate matrix inversions that involve adjacent matrices. While it is conceivable to develop a combined GRM integrating the SLERP, LUM and g2o algorithms, such an approach would negate our GRM’s primary advantage: its linear, parameter-free, and iteration-free nature.

## 5. Conclusions

In this work, we introduce two significant contributions to the field of 3D point cloud registration. Our first contribution, a refinement of existing methodologies, consists of a coarse-to-fine registration method combining FGR and multiscale-implemented GICP models. This approach particularly enhances the handling of point clouds acquired through TLS. The second and more substantial contribution is the development of a fully linear Global Refinement Model (GRM). This model stands out as it is free of parameters and iterations and effectively corrects drift in pose circuits. We successfully tested the first model on two TLS-derived point cloud datasets, registering 49 cloud pairs with precision. Moreover, this model adeptly reconstructed a complex circuit from the NCLT dataset, consisting of point clouds obtained via MLS. Our second model, SLERP+LUM, underwent comparative testing with other models on this dataset. It demonstrated superior performance in terms of execution time and drift correction efficiency in closed circuits. Our research further concludes that the multiscale GICP implementation notably enhances registration accuracy, particularly in datasets with varied point cloud densities, proving crucial for diverse real-world applications.

The GRM model’s lack of parameters and iterations not only simplifies its usage but also increases its adaptability across different datasets without extensive pre-configuration.

The superior execution time of the SLERP+LUM model opens possibilities for real-time SLAM applications, marking a significant advancement in practical implementations in dynamic settings.

## Figures and Tables

**Figure 1 sensors-24-05112-f001:**
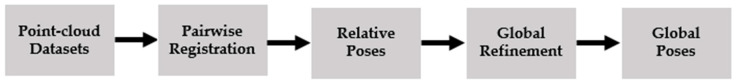
Core modules of our proposed 3D reconstruction method.

**Figure 2 sensors-24-05112-f002:**
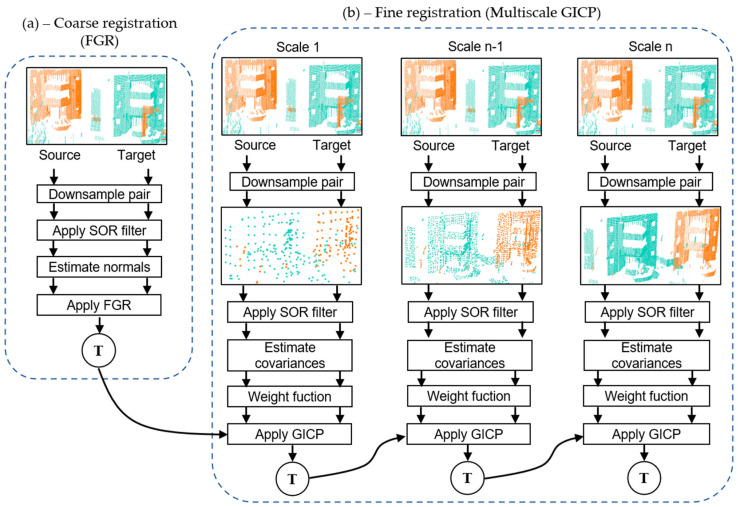
(**a**) Coarse registration steps using FGR. (**b**) Fine registration steps using Multiscale GICP. Downsampling, filtering, and normal estimation are standard pre-process steps in all point cloud pairs before registration; in FGR, it is necessary for the FPFH descriptor, and in GICP, it is necessary for the covariance estimation. Multiscale GICP also uses a weight function based on the L1 metric to penalize incorrect matches. The arrows show the point clouds through the modules of the coarse-to-fine FGR+M-GICP registration.

**Figure 3 sensors-24-05112-f003:**
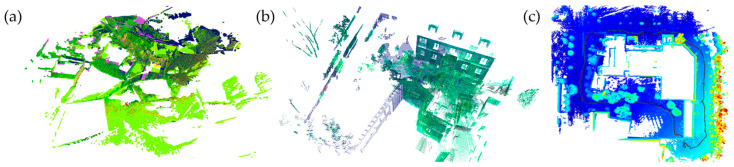
(**a**) View of the Courtyard dataset. (**b**) View of the Façade dataset. (**c**) View of the NCLT dataset.

**Figure 4 sensors-24-05112-f004:**
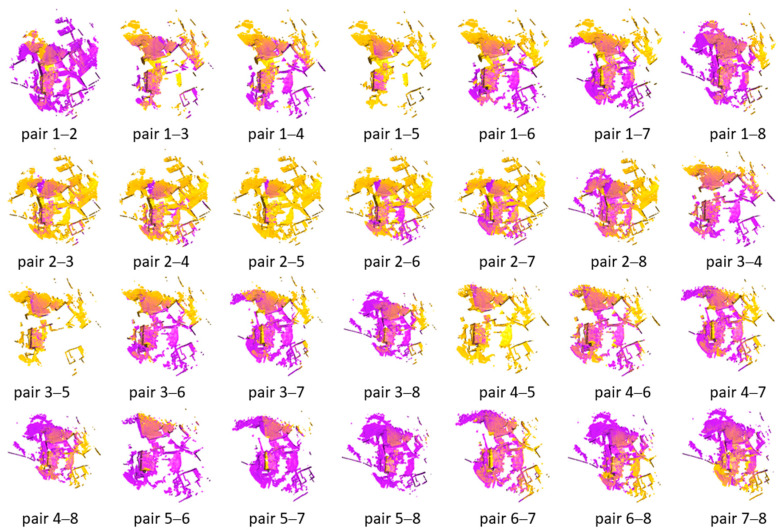
Visual inspection of the registration of all pairs in the Courtyard dataset by the proposed model FGR+M-GICP. The target cloud is painted magenta, while the source cloud is painted yellow.

**Figure 5 sensors-24-05112-f005:**
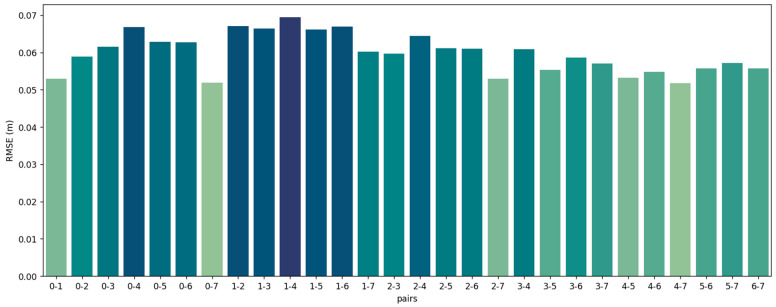
RMSE of all the correspondences in the overlap area of each point cloud pair from the dataset Courtyard.

**Figure 6 sensors-24-05112-f006:**
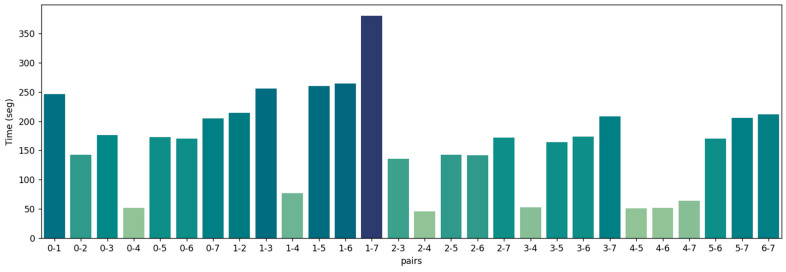
Registration time for all point cloud pairs of the Courtyard dataset. Mean time: 169 s.

**Figure 7 sensors-24-05112-f007:**
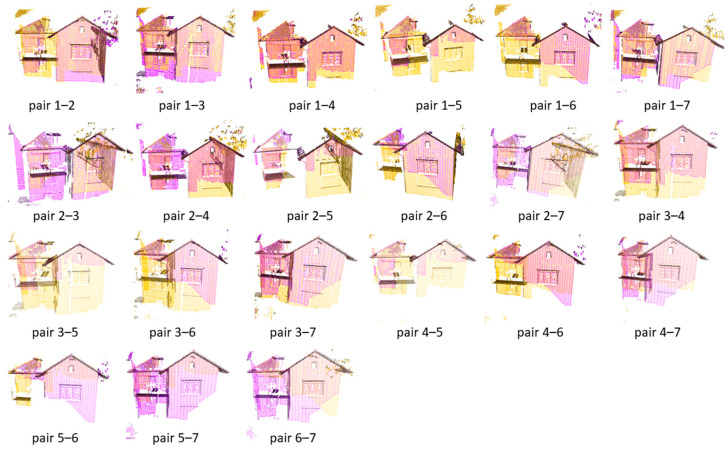
Visual inspection of the registration of all pairs in the Facade dataset by the proposed model FGR+M-GICP. The target cloud is painted magenta, while the source cloud is painted yellow.

**Figure 8 sensors-24-05112-f008:**
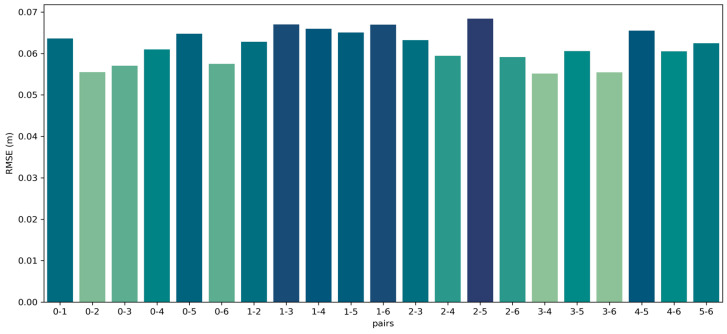
RMSE of all the correspondences in the overlap area of each point cloud pair from the Facade dataset.

**Figure 9 sensors-24-05112-f009:**
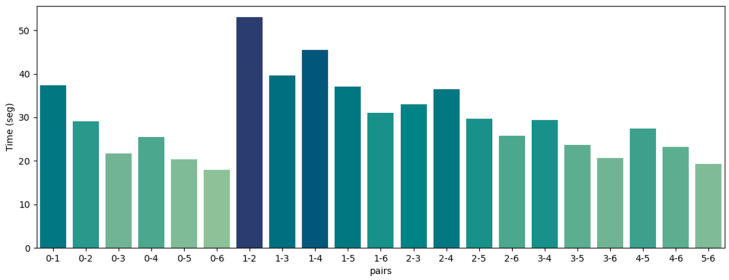
Registration time for all point cloud pairs of the Facade dataset. Mean time: 29.2 s.

**Figure 10 sensors-24-05112-f010:**
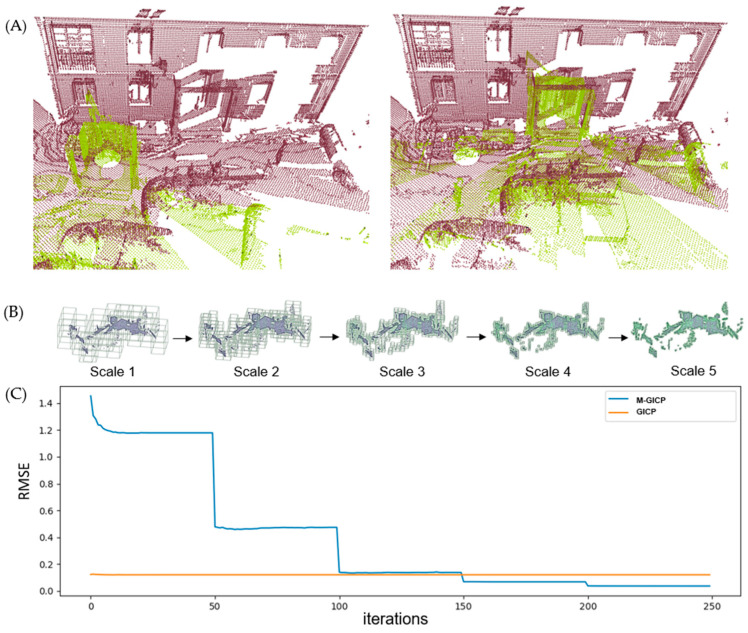
(**A**) Pair of point clouds 1–2 from the Facade dataset after registration with GICP Vanilla and M-GICP. (**B**) Multiscale downsampling voxels used in M-GICP. (**C**) RMSE over 250 iterations with 50 iterations per scale. We present a video animation of M-GICP in the Appendix A.

**Figure 11 sensors-24-05112-f011:**
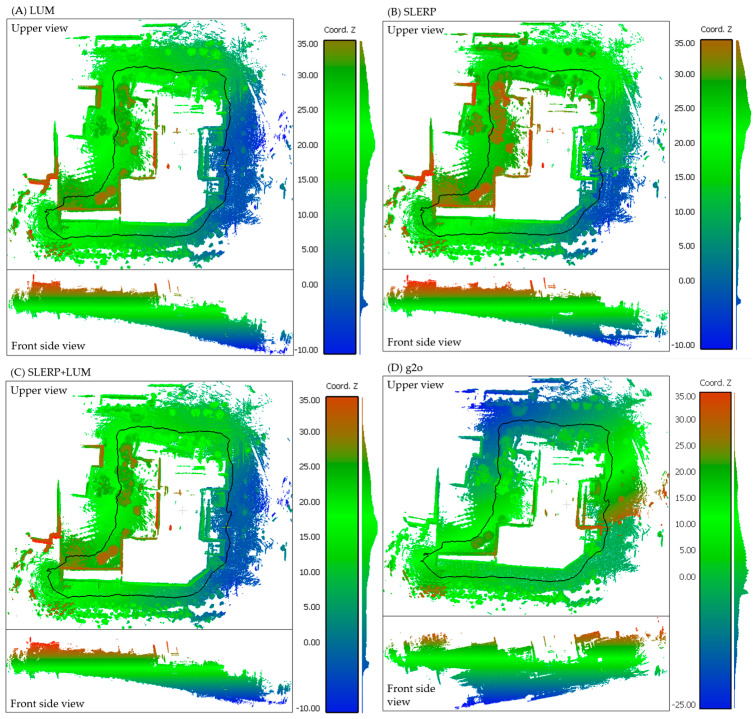
(**A**) LUM. (**B**) SLERP. (**C**) SLERP+LUM. (**D**) g2o. Drift correction in the NCLT dataset. For visualization purposes, the trajectory line has been smoothed. The color gradient shows elevation (Z coordinate) in an arbitrary coordinate system. The histogram at far right of each subfigure shows the distribution of points on the Z axis. The black line is the trajectory of the platform. We present a video animation of the circuit reconstruction in the Appendix A.

**Figure 12 sensors-24-05112-f012:**
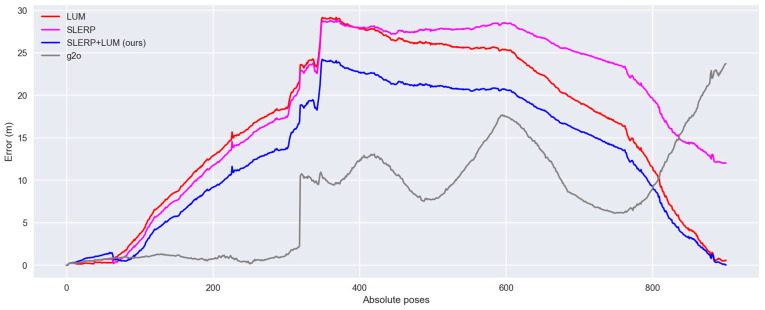
Pose error of each GRM. Line colors: red = LUM, magenta = SLERP, blue = SLERP+LUM, gray = g2o.

**Figure 13 sensors-24-05112-f013:**
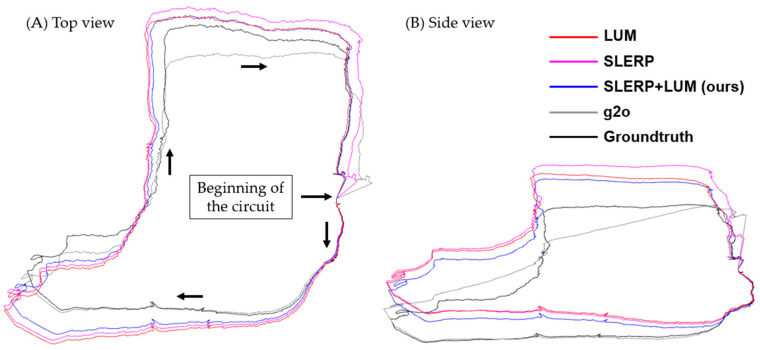
Estimated trajectories with each GRM and groundtruth. (**A**) Top view. (**B**) Side View. Line colors: red = LUM, magenta = SLERP, blue = SLERP+LUM, gray = g2o, black = groundtruth.

**Figure 14 sensors-24-05112-f014:**
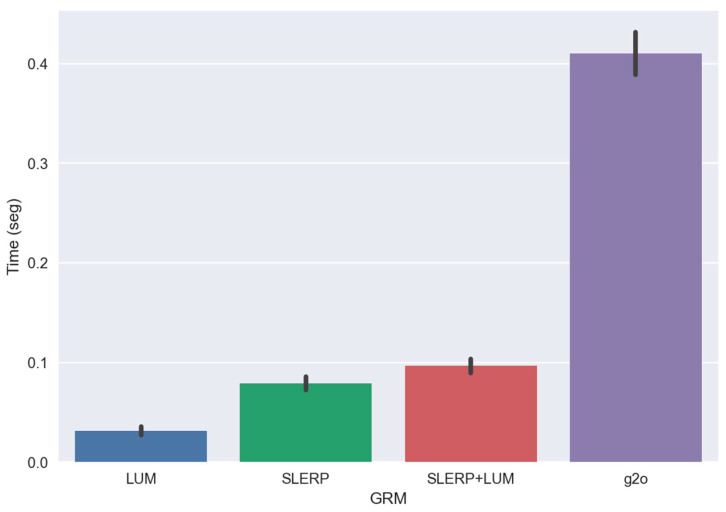
Estimated time of execution of each GRM. Median and interquartile deviation of 100 executions times.

**Table 1 sensors-24-05112-t001:** Parameters of FGR and Multiscale-GICP. knn= k-nearest-neighbors.

Parameter	Value	Description
Voxel size (VS)	VS=0.1 for FGR $VS=\left[0.5,0.4,0.3,0.2,0.1\right]$ for each scale of M-GICP	Used to reduce the number of points and uniformize the density along the point cloud pair
Maximum Correspondence Distance (MCD)	MCD=2 × VS for FGR $MCD=\left[3,2.5,2.0,1.5,1.0\right]$ × VS for each scale of M-GICP	Maximum distance to search for correspondences between source and target point cloud
Standard deviation multiplier $\left(\alpha \right)$ of the SOR filter	α=1.0 for FGR $\alpha =\left[0.6,0.8,1.0,1.2,1.4\right]$ for each scale of M-GICP	Scales the standard deviation of the SOR filter, making it less aggressive across scales
Neighborhood of the SOR filter ${\left(knn\right)}_{SOR}$	${knn}_{SOR}=30$ for FGR ${knn}_{SOR}=\left[64,32,16,8,4\right]$ for each scale of M-GICP	Number of neighbors used by the SOR filter
Neighborhood for normal estimation ${\left(knn,r\right)}_{MVC}$	${knn}_{MVC}=20,r_{MVC}=2$ × VS for FGR ${knn}_{MVC}=20,r_{MVC}=2$ × VS for all scales of M-GICP	Neighborhood with a hybrid approach that can use a maximum radius or a knn value.
Neighborhood for FPFH estimation ${\left(knn,r\right)}_{FPFH}$	${knn}_{FPFH}=200,$ $r_{FPFH}=10$ × VS	Neighborhood with a hybrid approach that can use a maximum radius or a knn value.

## Data Availability

TLS datasets can be downloaded at: “https://prs.igp.ethz.ch/research/completed_projects/automatic_registration_of_point_clouds.html”, accessed on 10 January 2024. The NCLT dataset can be downloaded at: “https://robots.engin.umich.edu/nclt”, accessed on 10 January 2024. All the code to reproduce results can be downloaded from the Github author account: “https://github.com/RubensBenevides/Point-Cloud-Registration-with-Global-Refinement”, accessed on 10 January 2024.

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
