# Peer review of "Advancing Global Pose Refinement: A Linear, Parameter-Free Model for Closed Circuits via Quaternion Interpolation"

_sensors, 2024, doi:10.3390/s24165112_

Round 1

Reviewer 1 Report

Comments and Suggestions for Authors

Global pose refGlobal pose refinement is a significant challenge within Simultaneous Localization and Mapping (SLAM) frameworks. In response to this challenge, the authors propose a linear, parameter-free model, which uses a closed circuit for global trajectory corrections. This paper presents two novel contributions to the field of LIDAR-SLAM. The first focuses on the LIDAR-SLAM front end, proposing a fully automatic method for registering multiple 3D point clouds using enhanced off-the-shelf methods. The second contribution targets the SLAM back-end, introducing a unique global trajectory optimization technique. It provides new insight into the application of pose refinement on LIDAR-based SLAM systems. Still, there are some scientific questions the authors should consider further.

1. The method seems to be a combination of some previous works. Moreover, the authors should emphasize their core modules.

2. I have strong concerns about the dataset. The samples are few and highly similar, which makes the method prone to overfitting.

3. In the experimental analysis section, parameter analysis needs to be strengthened.

4. Time analysis is also important, and I hope the author can have further discussion.

5. The authors propose a coarse-to-fine approach that combines the FGR and Generalized Iterative Closest Point (GICP) in a multiscale manner, I don't quite understand the motivation behind the author's design.

6. The image quality should be improved. The drawing of some flowcharts is relatively unprofessional, and the color matching is relatively abrupt during the reading process. I hope the author can improve the drawing of flow charts, pipelints and other images in the future.

Reviewer 2 Report

Comments and Suggestions for Authors

My comments are marked in the manuscript. My scanner clipped some of my handwritten notes. The bottom line of page is "on the last two lines". The middle word on the top line of page 9 is "confusion", not "contusion". The bottom line of page 9 is "should be the reciprocals of the variances, so I think you mean w = 1/RMSE."

Comments on the Quality of English Language

Minor corrections of English grammar are marked on the manuscript.

Round 2

Reviewer 1 Report

Comments and Suggestions for Authors

I have no further comment. 

Author Response

Thank you for the reviews and comments.

Reviewer 2 Report

Comments and Suggestions for Authors

Several comments and suggestions are marked on the scanned manuscript. I only scanned pages with comments. The handwritten word clipped at the right margin of page 16 is "image".

I graded "Are the conclusions supported by the results?" as "Can be improved" because I am still not convinced by the words in lines 546 - 548 that your model is "qualitatively" superior. Are you referring to the qualitative difference in the height colors in figure 11D? I think that these differences are incorporated into the error graphs in figure 12, which in my mind still show that the g20 method is superior, since the sum over all 901 poses of [(SLERP+LUM) error - g20 error] is positive. If you actually have ground truth information for the camera orientation, graphs of orientation error might also be useful, especially since your innovations for quaternion interpolation are for the orientations.

Comments on the Quality of English Language

Minor corrections in English are noted on the scanned marked manuscript.
